# The Lymphatic Endothelium in the Context of Radioimmuno-Oncology

**DOI:** 10.3390/cancers15010021

**Published:** 2022-12-20

**Authors:** Lucía Suárez, María E. Rodríguez-Ruiz, Ana Rouzaut

**Affiliations:** 1Department of Biochemistry and Genetics, University of Navarra, 31008 Pamplona, Spain; 2Division of Immunology and Immunotherapy, Center for Applied Medical Research, Universidad de Navarra, 31008 Pamplona, Spain; 3Department of Radiation Oncology, University Clinic of Navarra, 31008 Pamplona, Spain; 4Navarra Institute for Health Research (IDISNA), 31008 Pamplona, Spain

**Keywords:** lymphatic system, radioimmuno-oncology, radioresistance, vasculature, lymphedema

## Abstract

**Simple Summary:**

The introduction of immunotherapy within the usual strategies for cancer treatment has meant an enormous advance in the survival of patients with very restrictive prognoses. However, a high percentage of patients still cannot benefit from it. The lack of response to immunotherapy is mainly due to resistance from the tumor stroma itself. The lymphatic endothelium constitutes a permeable vascular system from the stroma specially dedicated to balancing tissue homeostasis and leukocyte emigration. Most solid carcinomas present lymphatic microvasculature that participates in tumor progression ambivalently. On one hand, it facilitates the transit of immune cells toward the lymph nodes and therefore promotes the antitumor response; on the other hand, they constitute an accessible route through which tumor cells metastasize to the lymph nodes. Due to this double function, it is not easy to establish strategies that “educate” the lymphatic endothelium only in its antitumor function without compromising its participation in the immune response. In this review, we study how combinations of radiotherapy and immunotherapy can modulate lymphatic function to use them in therapeutic approaches.

**Abstract:**

The study of lymphatic tumor vasculature has been gaining interest in the context of cancer immunotherapy. These vessels constitute conduits for immune cells’ transit toward the lymph nodes, and they endow tumors with routes to metastasize to the lymph nodes and, from them, toward distant sites. In addition, this vasculature participates in the modulation of the immune response directly through the interaction with tumor-infiltrating leukocytes and indirectly through the secretion of cytokines and chemokines that attract leukocytes and tumor cells. Radiotherapy constitutes the therapeutic option for more than 50% of solid tumors. Besides impacting transformed cells, RT affects stromal cells such as endothelial and immune cells. Mature lymphatic endothelial cells are resistant to RT, but we do not know to what extent RT may affect tumor-aberrant lymphatics. RT compromises lymphatic integrity and functionality, and it is a risk factor to the onset of lymphedema, a condition characterized by deficient lymphatic drainage and compromised tissue homeostasis. This review aims to provide evidence of RT’s effects on tumor vessels, particularly on lymphatic endothelial cell physiology and immune properties. We will also explore the therapeutic options available so far to modulate signaling through lymphatic endothelial cell receptors and their repercussions on tumor immune cells in the context of cancer. There is a need for careful consideration of the RT dosage to come to terms with the participation of the lymphatic vasculature in anti-tumor response. Here, we provide new approaches to enhance the contribution of the lymphatic endothelium to radioimmuno-oncology.

## 1. Introduction

Radiotherapy is a technique based on the use of high-energy accelerated particles to damage the DNA of irradiated cells, preventing their replication and causing their death. It has been used for more than a century as a treatment for cancer to eradicate tumors, reduce recurrence or as palliative treatment. Nevertheless, sensitivity to irradiation differs from subject to subject: radiation dosage, fractioning irradiated tissue and volume of irradiation. In addition, there are some specific individual factors such as age, sex, lifestyle or genetics and epigenetics affecting radiosensitivity [1].

Mechanisms to restore radiation damage appear in tumors as they evolve, making them resistant to radiotherapy. These mechanisms are multifactorial and may be a result of intrinsic conditions, such as alterations in the DNA repair machinery, or they may emanate from tumor microenvironmental conditions such as hypoxia [2]. The tumor microenvironment plays a prominent role in counteracting the effects of radiation. For example, dysfunctional endothelial vessels impair oxygen access to the tumor and therefore lessen the amount of radical oxygen species produced as a consequence of radiation [3]. This hypoxic state promotes further infiltration from myeloid-derived suppressor cells (MDSC). Importantly, defective vasculature impairs immune cell migration into the affected areas to phagocyte radiation-induced tumor antigens and present them in the lymph nodes or cross-present them to a tumor cross-presenting DCs [4].

On the other side, the immune system constitutes part of the stromal compartment that may modulate the radiosensitivity of tumors. In fact, current treatment strategies combine radiotherapy with other therapies directed to harness stromal-derived resistances [5]. RT can induce antitumor immune response through the activation of the IFNγ-STING pathway. However, there are instances in which RT-induced immune mechanisms contribute to immune suppression through the upregulation of PDL-1 in tumor and immune cells by the recruitment of suppressive cells and through the induction of the secretion of proinflammatory cytokines and chemokines such as transforming growth factor β (TGFβ), chemokine CC motif ligand 2 (CCL2) or colony-stimulating factor 1 (CSF-1), among others by stromal cells [6,7,8].

Immunotherapy has revolutionized cancer treatment and is considered an integral piece of multimodal therapies [9]. It consists in the (re)awakening of self-defenses against transformed cells (i.e., pressing the gas pedal) or the elimination of the mechanisms that inhibit an adequate response (i.e., releasing the brakes) [10]. Current efforts to induce potent antitumor immune responses can be summarized in three categories: use of monoclonal antibodies that block or activate immune-signaling receptors, use of vaccines against tumor-specific antigens or introduction of tumor-directed vectors that express immunomodulatory cytokines and, lastly, the administration of immune cells engineered to strongly and specifically recognize tumor antigens to mount a comprehensive immune response. The use of immune checkpoint inhibitors (ICIs), such as cytotoxic T-lymphocyte-associated protein 4 (CTLA-4), programmed cell death 1 (PD-1) and its ligand (PDL1), constitutes the spearhead of the immense number of molecules being currently tested in clinical trials [11]. Unfortunately, only 30–40% of patients benefit from immunotherapy, and even fewer achieve durable responses [12]. Defects in the immune response to therapy can be attributed to any of the steps of the immunity cycle as described by Chen and Mellman [13]: release of tumor antigens, antigen presentation and effector cell activation in the lymph nodes, effective infiltration of effector cells into tumor stroma and overcoming immunosuppression. The vascular system plays a significant role in homeostasis of this virtuous cycle and can be modulated in order to improve immune response against tumors; immune cells ingress in the tumor through the blood vasculature and leave it mostly through the lymphatic vessels [14,15]. Although the effect of irradiation on blood vessels has been extensively studied, its effect on the lymphatic system is not well-described. The lymphatic system plays a fundamental role both in the migration and activation of immune cells, but it also contributes to metastasis. Taking into account both the beneficial and detrimental effects of RT on the immune compartment and on the endothelium per se, care should be taken when considering the modification of the vascular system as a new tool to improve immune responses to radioimmunotherapy of cancer. In this review, we will be focused on the effect of irradiation on the lymphatic endothelium, and we will describe possible benefits of targeting it together with radiotherapy.

## 2. Lymphatic Vessel Biology

The lymphatic system constitutes a unidirectional vascular network consisting of lymphatic capillaries, collecting vessels and lymph nodes. It is essential in the maintenance of fluid homeostasis as well as in the absorption of dietary lipids and the transport of immune cells and soluble antigens from peripheral tissues to lymph nodes [16]. The lymphatic vascular network begins in the initial lymphatic capillaries, which drain into pre-collecting vessels and finally terminates in collecting lymphatic vessels endowed with a continuous basement membrane, smooth muscle cells and valves to prevent retrograde lymph flow [17].

The lymphatic vessels are present in almost all tissues except the bone marrow. Lymphatic networks have a fractal geometric organization, which allows smaller, more distal, blind-ended vessels to cover a large surface area within the tissues to absorb fluids, serving as the site of lymph formation. Thus, lymphatic networks generally begin with lymphatic capillaries that serve as the portal of entry for interstitial fluid absorption, and they are frequently found close to the local microvasculature, following the arterial network. In many cases, pre-collector lymphatics with one-way valves are present within the tissue, whereas collecting lymphatic vessels are located inside or at exit points of an organ [18]. Some organs, such as the mesentery, present many pumping collecting lymphatics, whereas those that contract frequently, such as the heart, seem to require less intrinsic pumping from the collecting lymphatics.

Despite differences in the structure of the lymphatic network between organs, the ultrastructure of the lymphatic vessels and the protein markers that identify the lymphatic endothelium is similar in most organs [18]. Among these markers are podoplanin, a transmembrane glycoprotein expressed in the lymphatic endothelium critical for embryonic lymphatic development, LYVE-1, a hyaluronic acid receptor expressed predominantly in lymphatic vessels, prox-1, an endothelial cell-specific transcription factor for lymphatics and the vascular endothelial growth factor receptor 3 (VEGFR3).

The initial lymphatic capillaries’ morphology facilitates migratory transit since they present an interrupted basement membrane composed of collagen IV, laminins, perlecan, nidogen [19] and a loose, button-like distribution of their inter-endothelial junctional complexes [16]. In addition, the initial lymphatic capillaries connect to the extracellular matrix by anchoring filaments that pull open the vessel walls at times of high interstitial pressure, causing these junctions to open to enhance fluid and particle absorption [20].

Lymphangiogenesis is a dynamic process that occurs during embryogenesis. In adult tissues, it only occurs during the female menstrual cycle and wound healing. In these last cases, it develops in parallel to angiogenesis [21] and primarily by sprouting from pre-existing vessels [22,23]. In some circumstances, bone marrow-derived cells, such as macrophages, can transdifferentiate into lymphatic endothelial cells and contribute to new lymphatic sprouts [24]. Lymphatic vessel growth accompanies several pathologies including cancer, chronic inflammation and transplant rejection [21].

The main factors governing lymphangiogenesis are the vascular endothelial growth factors VEGFC and VEGFD that bind to the receptor VEGFR3, a tyrosine kinase receptor that is expressed primarily in lymphatic vessels [25,26]. Activation of the VEGFR3 receptor leads to phosphorylation of the serine/threonine kinases AKT and ERK, which promote lymphatic endothelial cell proliferation, migration and survival [17,27,28]. VEGFD is similar to VEGFC, albeit dispensable for lymphangiogenesis in developing mammals [29]. VEGFD absence leads to lymphatic vessels of smaller caliber in the skin [30]. The vascular endothelial growth factor 2 (VEGFR2) is expressed at low levels on lymphatic endothelial cells, and the expression of its ligand, the vascular endothelial growth factor A (VEGFA), can stimulate lymphatic vessel expansion but not new vessel sprouting [31,32]. It has been shown that VEGFR3 is essential to control the expression of VEGFR2 and therefore modulates VEGFA/VEGFR2 signaling and, in turn, the permeability of blood vessels. Other growth factors, such as fibroblast growth factor (FGF), epidermal growth factor (EGF), hepatocyte growth factor (HGF) or platelet-derived growth factor (PDGF), intervene in lymphatic growth through the activation of tyrosine kinase receptors expressed in lymphatic vascular cells [33].

In parallel to their involvement in tissue homeostasis, the lymphatic vessels constitute conduits for immune cell trafficking. Lymphatic endothelial cells (LECs) secrete leukocyte chemoattractants, mainly chemokine (C-C motif) ligand 21 (CCL21), that deposits on positively charged extracellular matrix molecules in the form of concentration gradients. These gradients usher in C-C chemokine receptor type 7 (CCR7), expressing leukocytes towards the lymph nodes [34]. In healthy tissues, the transit of leukocytes through the lymphatic endothelium appears to be passive [35], and leukocytes access the lymphatic endothelium through the discontinuities present in the basement membrane and inter-endothelial contacts [36].

In contrast, under inflammation, leukocyte trafficking through the lymphatic endothelium is regulated by integrins expressed on the surface of leukocytes and their specific ligands expressed on the endothelial surface. The most relevant integrin–ligand pairs in this process are those made up of the integrin lymphocyte function-associated antigen 1 (LFA-1 or αvβ2), its ligand, intercellular Adhesion Molecule 1 (ICAM-1), the integrin very late antigen-4 (VLA-4 or α4β1) and its ligand, vascular cell adhesion protein 1 VCAM-1 [36,37].

Once in the lymph node, chemokine gradients will guide DCs and other immune cells toward the parenchyma and further into the sinus [38]. The local signaling microenvironment determines the type and intensity of the immune response in the lymph nodes. Although most peripheral antigens reach the lymph node parenchyma on antigen-presenting cells (APC), minor antigens (less than 70 kDa) can reach the sinus on the lymph well before the influx of antigen-loaded APC. These antigens transit across the lymph node-LECs through transendothelial channels, reticular conduits and active transcytosis [39].

The lymphatic vessels also intervene in the modulation of the immune response. Lymph node LECs are highly phagocytic and endocytic. For example, it has been shown how these endothelial cells can archive and further exchange viral antigens to migratory dendritic cells (DCs) in the context of vaccination or viral infection [40]. In addition, lymph node LECs contribute to peripheral tolerance to autoantigens through the constitutive expression of major histocompatibility complex (MHC) class I molecules and the presentation of autoantigens in the absence of costimulatory factors, leading to the elimination of autoreactive T cells [41]. Tolerance to peripheral antigens in the node is also possible because lymphatic endothelial cells express PD-L1, which limits CD8+ T cell activation. The melanocyte protein tyrosinase is an example of a peripheral tissue autoantigen presented by lymphatic endothelial cells. If the lymph node endothelial cells lose PD-L1 expression, autoimmune vitiligo is generated [42].

Moreover, lymphatic endothelial cells can induce anergic CD4 T cells presenting antigens in MHC class II either endogenously expressed or through autoantigen-loaded MHC class II molecules acquired from dendritic cell-derived exosomes. Lymphatic endothelial cells also present antigens to CD4 T cells in a context deprived of costimulus signals and with high PD-L1, resulting in the education of CD4 T cells into an anergic state. If PD-L1 expression in the endothelium is inhibited or autoantigen–MHC class II complexes are not accepted by lymphatic endothelial cells, autoimmune reactions occur [43,44,45]. Of interest is how peripheral LECs also express high PD-L1 under some circumstances (i.e., high IFN-γ), which may modulate the immune response in the periphery. Lastly, lymphatic endothelial cells can directly prevent dendritic cell maturation through the secretion of immunosuppressive factors such as nitric oxide [46], indoleamine-pyrrole 2,3-dioxygenase (IDO) [47], or via ICAM mediated cell adhesion to macrophage-1 antigens (MAC-1) expressed on DC surfaces. MAC-1-dependent cell adhesion leads to decreased expression of the costimulus molecule CD86 by DCs and, thus, a reduced ability to activate effector T cells [48].

## 3. Lymphatic Vessel Biology in Cancer

Lymphangiogenesis is an early event in the natural history of cancer progression, and many patients will already have lymph node (LN) metastases on initial presentation [49]. Initially quiescent, tumor lymphatic vessels become lymphangiogenic in response to the expression of pro-lymphangiogenic factors, such as VEGFC produced by tumor and tumor stromal cells. This change translates, in most cases, into an outbreak of new lymphatic capillaries or dilation of the initial and collecting lymphatics already present in tumor tissue. Several angiogenic factors present in solid carcinomas, such as FGF, EGF, HGF, PDGF, angiopoietins and adrenomedullin, are also lymphangiogenic [50,51].

Tumor stromal reprogramming also contributes to lymphatic vessel expansion. Hypoxia can enhance VEGFC translation [52,53], and interstitial pressure increases tumor lymphangiogenesis in order to reduce tumor-associated edema [54]. Myeloid cells are an essential source of VEGFC in tumors. For example, tumor-associated macrophages (TAMs) secrete high amounts of VEGFC/D [55]. In addition, IL-1β from tumor-associated macrophages (TAMs) can directly stimulate LEC proliferation and migration [55]. Surprisingly, TAMs can also transdifferentiate into LECs under pathogenic conditions [56]. Other leukocyte-derived inflammatory cytokines, such as IL1β, TNFα or IL10, attract VEGFC-secreting leukocytes that amplify tumor lymphangiogenic activity [38,57]. Cancer-associated fibroblasts (CAFs) also contribute to lymphatic vessel proliferation and permeability through the secretion of VEGFC and other lymphangiogenic factors [58].

As it occurs in blood vessels, tumor lymphatic vessels present an immature and aberrant structure that can actively contribute to tumor metastasis [21,51]. For example, tumor VEGFC can enhance lymphatic endothelium permeability by disrupting the cadherin/β-catenin complex at intercellular junctions of LECs, facilitating tumor cell entrance towards the lymphatic vessels [59]. In fact, the expression of VEGFC is associated with increased metastasis in lymph nodes and distant organs. It is also associated with a worse prognosis in certain cancers, such as breast, lung or gastrointestinal tract [28,60,61]. In addition, blockage of VEGFR3 decreases peritumoral lymphangiogenesis and suppresses tumor metastasis in transgenic mice [62,63,64].

VEGFC accumulated in the tumor context induces tumor lymphatic vessels to release chemokines such as CCL19 and CCL21 that can attract tumor cells through their binding to CCR7 [65]. Other chemokines’ axes triggered by lymphatic endothelial cells are the CCL27/28 [66], CXCL12 [67,68] and CXCL10-CXCR3 axis described in colon cancer [69]. Interestingly, the amount of S1P secreted by LECs to control lymphocyte exit from LNs positively correlates with metastasis of cancer cells [70,71]. In addition, tumor cells can also arrest inside the lymphatic vessels while “in transit” to LNs [72]. Lastly, tumor-secreted VEGF-C and midkine ligands can accommodate the pre-metastatic niche by inducing lymph node lymphangiogenesis before metastatic seeding [73,74,75].

Therapies directed to modulate tumor lymphangiogenesis have gained interest recently. Different strategies using small-molecule inhibitors, blocking mAbs or VEGFC trap molecules have been developed that target VEGFC/VEGFR3 signaling [28]. The preclinical assays in a good number of animal models repeatedly manifested benefits in reducing tumor lymphatic vessels and metastases. For example, in preclinical models of breast cancer, treatment with VEGFR3 blocking antibodies or sVEGFR3 as a decoy receptor reduced chemotherapy-induced lymphatic metastases [63,76]. In addition, the small-molecule-specific VEGFR3 tyrosine kinase inhibitor SAR131675 significantly reduced the growth of the primary tumor, lung metastases and macrophage infiltration in a mouse model of breast carcinoma [77]. Although promising in the preclinical setting, the clinical benefits obtained were modest; in fact, it has been reported how treatment with the anti-angiogenic molecule sunitinib induce VEGFC expression, lymphatic vessel density, and lymphatic metastases in renal carcinoma tumors [78,79]. Therefore, the combinations of antiangiogenic and antilymphangiogenic agents need to be cautiously measured.

Targeting the VEGFC/VEGFR3 pathway in combination with immunotherapy is being considered as a candidate strategy for treating cancer. Pre-clinical evidence demonstrates how VEGF-C-induced lymphangiogenesis potentiates the efficacy of anti-PD-1 therapy in heterotopic B16 melanoma in mice. In this case, as in others, VEGFC-mediated responses were dependent on signaling through the CCL21/CCR7 pathway. Importantly, the authors showed a positive correlation between serum VEGF-C levels and the presence of peptide-specific CD8+ T cells and prolonged progression-free survival in patients treated with anti-PD1 and anti-CTLA-4 checkpoint blockades [80]. Additional evidence shows the efficacy of a pro-lymphangiogenic vaccine in the induction of robust CD8+ T cell responses in the tumor and draining lymph nodes against melanoma tumors [81]. Evidence of a therapeutic opportunity in tumors different from melanoma is emerging. For example, it was recently shown in mice models how meningeal lymphatic vessels and VEGF-C are essential for the egress of CD8+ T cells from tumors and the subsequent priming in the lymph nodes to induce an efficient immune response against brain tumors [82,83]. All this evidence points to lymphatic vessels as a relevant system to be harmonized with immune therapy to optimize the results of cancer immunotherapy.

## 4. Effects of Ionizing Radiation on Tumor Niche

Despite the tremendous advance immunotherapy has brought to cancer treatment, immune checkpoint blockade (ICB) as a standalone therapy account only for 20% of the objective clinical responses with significant toxicities [84]. Thus, mounting pre-clinical and clinical efforts are dedicated to identifying combinations of immunotherapy with other therapies to improve patient survival with decreased morbidity. Combinations of immunotherapy with radiotherapy offer synergisms from RT-derived immunomodulatory effects [85,86,87]. Interestingly, while radiotherapy deters the growth of the irradiated primary tumors by a T cell-dependent mechanism, combinations with ICB (CTLA-4, PD-1) may induce the destruction of both the irradiated primary tumor and non-irradiated metastases through a still-discussed systemic effect called the abscopal effect, which is also cell-dependent.

Depending on the dose, radiotherapy can induce or suppress the immune response by direct action on the tumor itself or almost in any type of cell in the tumor niche [88]. In terms of in situ vaccination, higher radiation doses present more immune-stimulating effects than lower doses (<2 Gy), which induce anti-inflammatory and immune-suppressive effects [89].

Immunogenic cell death (ICD) is a prominent radiation-induced mechanism to modulate synergism with immunotherapy. In this case, specific antigens and alarmins are secreted with inflammatory cytokines, causing the activation of the innate and adaptive immune system by cross-presenting tumor antigens to CD8+ T cells by a subset of a specialized dendritic cell named Baft3 DC [90,91]. Radiation also contributes to increased tumor “visibility” for the immune system through the induction of MHC I expression in the tumor microenvironment [92] and the promotion of tumor regression of non-irradiated lesions [93] (Figure 1A).

However, with relative frequency, the tumor reappears after treatment due to the acquisition of resistance mechanisms, immunological evasion or through DNA repair. In fact, IFNβ production can be attenuated through the action of exonucleases, such as TREX-1, that activate the cGAS-STING pathway [94].

Hypoxia and growth factors secreted by the irradiated tumor stroma also contribute to resistance. On one hand, hypoxia limits the sensitivity of tumors to radiation, which preferentially kills well-oxygenated cells. In fact, cells irradiated in the absence of oxygen are two to threefold more radioresistant than well-oxygenated cells [95]. The cellular adaptation of the tumor to hypoxia is driven by the EGLN/HIF-1 axis that is active in different components of the tumor microenvironment [96], including tumor endothelial cells. HIF-1 activation post-irradiation protects the vasculature countering the oxidative stress caused by irradiation, leading to maintained tumor–vessel integrity and tumor perfusion [97]. Interestingly, it has been published recently how targeting of HIF-1α post radiotherapy improves antitumor immunotherapy and the efficacy of radiotherapy [98]. Growth factors such as TGFβ, secreted in response to RT by tumor-infiltrating fibroblasts and macrophages, promotes DNA repair and limits immune activation [99]. In addition, SDF1, CSF-1 and CCL2 attract the infiltration of myeloid suppressor cells and promote the neovascularization of tumors (Figure 1B). In this sense, the mobilization of inflammatory monocytes via the CCL2/CCR2 axis has been described as a negative prognostic factor in several cancers, including pancreatic cancer [100]. Tumor-infiltrating macrophages are bivalent depending on their pro-inflammatory (M1-like) or pro-tumoral (M2-like) phenotype. M2-like macrophages secrete immunosuppressive substances such as IL-10, VEGF, CCL2 or TGFβ [101]. Interestingly, it was described how M2-like macrophages can reverse their phenotype towards being M1-like under low-dose irradiation conditions (2 Gy) and favor vascular normalization, cytotoxic lymphocytic infiltration and improved antitumoral response [102]. Radiation also induces the expression of PD-L1 in the tumor stroma as a mechanism of acquired resistance induced by IFNγ produced by activated CD8+ T cells [103].

It is necessary to weigh the toxicity that radiotherapy can exert on healthy organs. For example, tumor-resident T cells are resistant to radiotherapy, while lymph node T lymphocytes are sensitive. In fact, radiation to the lymph node reduces therapeutic efficacy by the toxicity to tumor-specific lymphocytes [104]. In addition, lymphopenia caused by radiotherapy has been associated with recurrences and worse prognosis in triple-negative lung, cervical and breast cancers [105,106]. In the case of breast cancer, post-radiotherapy lymphopenia-associated recurrence appears to be mediated by the pro-metastatic activity of tumor-infiltrating macrophages that recruit new tumor cells to establish new metastatic niches [107].

## 5. Effects of Ionizing Radiation on the Vasculature

### 5.1. Effects on Healthy Endothelial Cells

Vascular endothelial injury is an important component of late radiation-induced morbidity and affects several tissues and organs, including the skin [108], kidney [109], lung [110], bowel [111] and heart. In fact, coronary diseases such as stroke could be prompted due to thrombosis and atherosclerosis initialization and acceleration post-radiation treatment [112,113]. In this sense, it is established how, depending on the RT protocol (the number of fractions, dose rate and the total dose of radiation applied), the effects on the status and functionality of the vasculature differ [114]. Most RT treatments use external beams based on photon radiation, concretely X-rays, while other approaches use internal radioactive sources of gamma rays, such as those used for brachytherapy, systemic RT with targeted radionuclides or high linear transfer energy (LET). In terms of effects on the vascular system, the effects on the targeted tissue are more dependent on the amount of energy transferred than from the source of energy per se.

Microvascular damage is an essential component of radiation-induced late morbidity and affects most tissues and organs, including the kidney, lungs, heart, bowel and skin [115]. RT, mainly when used at high doses, induces increments in endothelial permeability, its detachment of the basement membrane and cell apoptosis or senescence. In addition, it causes a decrease in the number of pericytes, all of them crucial for the regulation of immune cell infiltration [116,117]. If damage to the endothelial layer becomes chronic, a state of inflammation, fibrosis, and hypoxia that favor thrombosis, atherosclerosis, necrosis and even damage to the heart of the irradiated tissue may occur [118,119,120].

At the cellular level, exposure of healthy tissue to ionizing radiation at >10 Gy induces the accumulation of senescent irradiated endothelial cells in tissues that secrete inflammatory mediators and proteases that contribute to chronic inflammation and to disruption of the vascular structure [121]. Pro-inflammatory cytokines such as TNFα or IL-6 released after RT lead, in some instances, to vascular activation, including the increased expression of the adhesion receptors ICAM-1, VCAM-1 and E-Selectin or to the modification of the glycosylation pattern on the vascular surface [117,122,123]. These changes contribute to immune cell infiltration after RT and chronic inflammation when exacerbated. Of note, it was described how ICAM-1 KO mice show less-severe pneumonitis after irradiation [124]. Depending on the dose, the incremented vascular permeability can be integrin-independent and a consequence of temporal deconstruction of the junctional complexes between endothelial cells through Rho kinase-mediated alterations of the cytoskeletal structure [125]. For example, there are instances in which fractionated RT augments vessel permeability for extended periods by incremented ZO-1 and ICAM-1 expression on the endothelial surface [126]. Recent research points to high linear transfer energy (LET), such as neutron transfer, to present less pro-fibrotic and more pro-immunogenic properties than gamma ray- and X-ray-based RT approaches [127].

### 5.2. Effects on Tumor Endothelial Cells

The endothelial cells of the tumors present a limited capacity to undergo RT-driven senescence, most probably associated with their proliferative status. The overproduction of angiogenic stimuli in the tumor stroma produces leaky and tortuous microvascular architecture that associates with defective tissue oxygenation and hypoxic areas in the tumor [128]. Radiotherapy-induced changes in tumor vasculature also depend on the type, location and stage of the tumor itself [114]. Depending on the origin of the tumor vasculature (i.e., by angiogenesis, vasculogenesis or vessel cooptation), tumor vessels may lack basement membrane and pericyte coverage, making them more permeable, leaky and invasive or more radiosensitive than surrounding normal tissue vessels [129]. This lower oxygenation leads to an inadequate response to RT in tumors [130]. In this sense, there exists controversy regarding the positive or negative contribution of endothelial cell death to RT derived tumor responses [131,132].

At this point of the debate, it seems clear that depending on the dose and frequency of irradiation, the response of this cell type is different. Evidence obtained in pre-clinical experiments shows how irradiation with >10 Gy induces severe vascular damage and results in reduced blood perfusion. In comparison, administration of a single fraction with doses lower than 5–10 Gy leads to temporal and mild changes in endothelial cell permeability through impairment of the inter-endothelial junctions [114]. In addition, radiotherapy induces a pro-angiogenic shift [133]. Specifically, stromal fibrosis and the induction of VEGFA from tumor cells promote tumor regrowth and angiogenesis [134]. Lastly, it has recently been described in brain tumors how exposure to small doses (<1 Gy) induces normalization of the blood vascular endothelium. Interestingly, myeloid cells are recruited to tumors partly through the interaction between the HIF-1-dependent stromal cell-derived factor-1 (SDF-1) and its receptor, CXCR4 [135].

Thus, at high doses of radiation, the endothelium can be dysfunctional and cause a lack of nutrients, hypoxia and a lower leukocyte infiltrate. On the contrary, if the endothelium receives doses small enough to normalize its phenotype without inducing angiogenesis, a greater leucocyte infiltration is achieved as a result of the expression of the integrin receptors ICAM-1 and VCAM-1 on the endothelial surface globally facilitating the activation of an effective immune response.

Consequently, it is necessary to optimize the radiotherapy doses received, considering their possible (or detrimental) effects on the endothelium. In this sense, in recent years, low-dose radiotherapy has been proposed in metastatic lesions as a way of reversing the immune desertification of tumors [136]. However, the results obtained in the clinical setting have still been modest [137] and re-emphasize the importance of exquisite design in terms of dose, frequency and duration of treatment. All of these aspects are summarized in Figure 2.

### 5.3. Effects on the Immune System

Endothelial cells of the tumor microenvironment are essential in immune exclusion and in the inhibition of lymphocyte activation, a status known as “endothelial anergy” that contributes to immunosuppression [138]. This is characterized by shallow expression of the leukocyte adhesion receptors E-selectin, VCAM-1 and ICAM-1 due to inhibition by pro-angiogenic factors [139]. As mentioned above, low irradiation can reverse anergy through transient activation of the immune response by endothelial normalization, increased expression of adhesion receptors, tissue oxygenation and the recruitment of tumor-specific T cells [140,141]. For example, radiotherapy induced the expression of the chemokine CXCL16 in the tumor microenvironment that may recruit CXCR6-positive effector CD8+ T cells [142]. Lastly, low-dosage RT may also increment the infiltration of dendritic cells and contribute to switching the phenotype of tumor-infiltrating suppressive macrophages to the iNOS/+M1 type. All these effects are at least partly mediated through the inhibition of the production of pro-angiogenic molecules [98,143]. In this sense, accumulation of TAMs in the vicinity of the microvasculature of irradiated tumors has been used as a therapeutic strategy to increment the success of the administration of drug-loaded nanoparticles loaded [144,145].

## 6. Effects of Radiotherapy on the Lymphatic Endothelium: Lymphedema

In the clinic, the lymphatic vessels are often included in the irradiated field and suffer the consequences of ionizing radiation, albeit at a different magnitude than their blood counterparts. Being the conduits in charge of the transit of antigen-presenting cells from the tumor to the lymph nodes to mount the antitumor response, it is surprising that relatively few studies address how ionizing radiation affects their structure and functionality.

Although the lymphatic vasculature is not covered by pericytes and present intermittent adhesion structures, they are suggested to be more resistant to radiotherapy than blood vessels. Several experiments support this notion. For example, in an analysis of the lymphatic vasculature in skin biopsies from breast cancer patients, the total amounts of lymphatic vessels were similar in irradiated and non-irradiated tissues. However, irradiated tissues presented lower numbers of small-caliber vessels, while there was an increment in the total amount of high-caliber lymphatics. Loss of the lymphatic microvasculature in the irradiated field was associated with higher TGFβ levels in the irradiated tissue, while VEGF-C secretion by tumor resident macrophages was suggested to be associated with the increments observed in high-caliber vessels [115]. In addition, it was demonstrated in mice how lymphatic endothelial cells of the small intestine and peri-tumoral areas were more resistant to radiation injury than blood vessels [146].

As it occurred with other tissues, the damage infringed on the lymphatic vasculature depends on radiation dosage. In mouse models for lung cancer that received ionizing radiation, it was demonstrated how radiated A549 and H1299 NSCLC cells secreted VEGFC in a dose-dependent manner that resulted in increase in MLVD in irradiated tumors with five fractions of 5 Gy when compared to non-irradiated mice. The same vessels present severe necrosis when mice were irradiated with five fractions of 10 Gy [147]. Proliferative vessels, such those present in tumors, are by nature more sensitive to irradiation than stable adult vasculature. For example, most tumors secrete lymphangiogenic factors; at least in vitro, the lymphangiogenic factor VEGF-C enhances the radiosensitivity of LECs [148].

In a study of high-dose irradiation of mouse lungs, Cui et al. reported a significant decrease in lymphatic vessels associated with radiation exposure in spite of presenting higher amounts of VEGFC- and VEGFD-positive alveolar macrophages. In addition, the fibrotic lesions in irradiated mouse lungs exhibited strong immunoreactivity for VEGFC [110].

It seems that lymphatic endothelial cells respond to ionizing radiation through stress-induced senescence. In line with this, in experiments performed in vitro, lymphatic endothelial cells exposed to single doses of 4, 8 or 12 Gy showed incremented senescence with doses with only a residual (8%) induction of apoptosis upon 15 Gy [149]. In this sense, depending on the vascular damage, the lymphatic vasculature mounts reconstructive responses through the secretion of growth factors and cytokines. The secretion of VEGF-C in tissue, promoted by COX2-dependent prostaglandins produced by TAMs and tumor cells under inflammation [150], has proven to promote the restructuration of the lymphatic vasculature [147]. In contrast, some authors did not find any increment in VEGFD or VEGFC expression, or that of their receptor VEGFR3, in radiation-resistant intestinal lymphatic endothelial cells in response to single-dose whole-body radiation in murine tumor xenograft models [146]. In this last case, other lymphangiogenic factors different from VEGFR3 ligands may be induced upon RT. Between them: VEGFA, whose secretion is increased after radiotherapy under hypoxic environment and can also induce lymphangiogenesis [151]; fibroblast growth factor 2 (FGF2) and platelet-derived growth factor beta (PDGFβ), which are induced in the tumor microenvironment after fractionated radiation in murine tumors and also have a lymphangiogenic role [152,153]. In contrast, RT can also induce anti-lymphangiogenic factors such as TGFβ, which is associated with radioresistance [154]. Therefore, there is still room to study the specific growth factors and circumstances that govern lymphatic endothelial cell sensitivity or resistance to radiation therapy.

Lymphedema is a common condition in patients with breast cancer and is characterized by a malfunction and destabilization of the lymphatic vessels. This debilitating condition appears as a consequence of damaged lymphatic vessels that lead to edema, fibrosis, inflammation and dysregulated adipogenesis. In consequence, there is marked swelling of the affected limb with all the associated morbidities.

There are two types of lymphedema: primary and secondary. Primary lymphoedema results from abnormal development and/or functioning of the lymphatic system. Mutations in numerous genes involved in the initial formation of lymphatic vessels (including valves) as well as in the growth and expansion of the lymphatic system are identified in about one-third of affected individuals [155]. Alterations of the VEGFR3 signaling pathway are a common feature of primary lymphedema, and supplementation with VEGFC-releasing patches or administration of HGF or ANG-2 improves the lymphedema condition, at least in animal models [156]. Secondary lymphedema, however, can be triggered in response to surgery or irradiation of lymphatic vessels. Unlike lymphatic vessels, lymph nodes are highly radiosensitive due to the fatty acid change and fibrosis experienced after being irradiated [157]. Therefore, the risk of developing lymphedema might be higher if lymph nodes have been resected or if the irradiated tumor has few or many nearby lymph nodes [158]. The risk of developing lymphedema increases exponentially with RT dosage and distance of the lymph node to the irradiated organ: the closer LNs are to the irradiated organ, the higher the effect [159].

In breast cancer, lymphedema affects 1 in 4 patients who undergo RT. As reviewed by Allam et al., RT increases the risk of lower extremity lymphedema by up to 40% in patients with gynecologic cancers. Furthermore, the RT delivery technique also affects lower extremity lymphedema rates [160]. In consequence, radiation therapy is one of the risk factors behind lymphatic damage adding to genetic [161,162] and pathophysiological factors, such as partial lymphatic resection during cancer surgery, infections [163], inflammation [164], venous diseases or obesity [165].

In this sense, Kwon et al. found in a mouse model that remodeling in the lymphatic endothelium was proportional to the dose of radiotherapy used (8 and 20 Gy) after popliteal lymphadenectomy. This remodeling encompassed increased vessel dilation and abnormal indocyanine green clearance and lymphatic reflux, with higher lymphatic vascular density but less contractility [166]. Similar data were obtained after irradiating the lymph node in mice (four daily doses of 8 Gy) that had or had not undergone lymphadenectomy by measuring lymphatic function one week, one month and six months after treatment [167]. In addition, measurement of lymphatic microvasculature density one year after 30–40 Gy irradiation of patients showed double the amount of small lymphatic vessels (diameter <10 μm) in irradiated patients. This event positively correlated with lymphedema, the number of macrophages and the expression of VEGFC measured two- and eight-weeks post-radiation [118]. All of these results demonstrate that the risk of suffering from lymphedema in patients with breast cancer undergoing surgery increases after RT, even more so if the axillary lymph nodes have been dissected [160].

Patients with lymphedema often display impaired immune function that predisposes them to various infections and may compromise the efficacy of immunotherapy [168,169]. For example, preclinical studies combining RT with anti-PD1 suggest that irradiation of draining LNs affects T cell infiltration in the primary tumor due to modification of the expression of intratumoral chemokines and the reduction of overall survival, so irradiation of these LNs should be avoided [170].

Recently, the importance of a functional lymphatic vasculature to modulate the efficacy of RT-derived immunity has been demonstrated in a mouse model of glioma [171]. In this report, the authors show how the absence of meningeal and cervical lymphatic vessels reduced the ratio of CD8+ Treg cells and dendritic cell trafficking to the lymph nodes. Interestingly, mice that overexpressed VEGFC survived longer than the control group after RT. The direct intervention of RT in the modulation of leukocyte transit was recently addressed. For instance, radiation dose- and time-dependent induction of ICAM-1 and VCAM-1 on LEC surfaces in primary human lymphatic endothelial cells, mouse transplanting tumor models and in pre- and post-radiation patient samples have been described with an effect that persisted more than a week as a way to increment leukocyte transit in the context of radiation therapy [122].

## 7. Future Perspectives Modeling Lymphatics for Immunotherapy

The function performed by the lymphatic vessels in the tumor context is analogous to the philosophical principle of Yin and Yang. On one hand, the lymphatic vasculature contributes to tumor progression through lymphatic metastases and lymph node suppression (Yin). On the other hand, it offers portals to usher dendritic cells and tumor-specific lymphocytes toward the lymph nodes to mount an effective immune response (Yang). In addition, the same cytokines and growth factors govern these apparent opposing functions (i.e., CCL21 attracts both dendritic and tumor cells toward the lymphatic vasculature) (Figure 3).

As occurs in most physiological processes, the outcome of a specific response should be attributable to a complex balance between the cytokines, growth factors and adhesive contacts produced by tumor and stromal cells in a particular moment and under specific circumstances. Consequently, the key to harness lymphatic biology in this context will come from a holistic understanding of the tumor environment. The heterogeneous roles played by lymphatic vessels complicate their translation into the clinic, as treatments should impede lymphangiogenesis while promoting priming of the antitumor immune response at once.

In this sense, the addition of radiotherapy to treatment combinations must be done with some caveats: (i) it should be administered at doses that maintain lymphatic integrity to allow correct DC migration to the lymph nodes by using low-dose radiotherapy regimes (<10 Gy) or adding growth factors that help to reduce endothelial toxicity, such as bFGF; (ii) at the same time, the production of immune suppressive factors such as TGFβ [172] should be kept to a minimum through the concomitant administration of TGFβ-specific inhibitors or CD40 agonists that can reprogram the tumor microenvironment; (iii) radiotherapy should be administered before immunotherapy with time enough to allow tumors to recover from temporal lymphopenia; (iv) especial attention should be driven to spare the adjacent lymph nodes, whose functionality have proven to be essential for proper antitumor response in response to immunotherapy [49].

In sum, the lymphatic vasculature constitutes an attractive target to educate the tumor environment to benefit the patients. Due to its complex nature, research on the endothelium has suffered lagged attention, but sustained research into the molecular pathways that govern its immunomodulatory capabilities as well as cell transit across it may help us to master it for therapeutics.

## Figures and Tables

**Figure 1 cancers-15-00021-f001:**
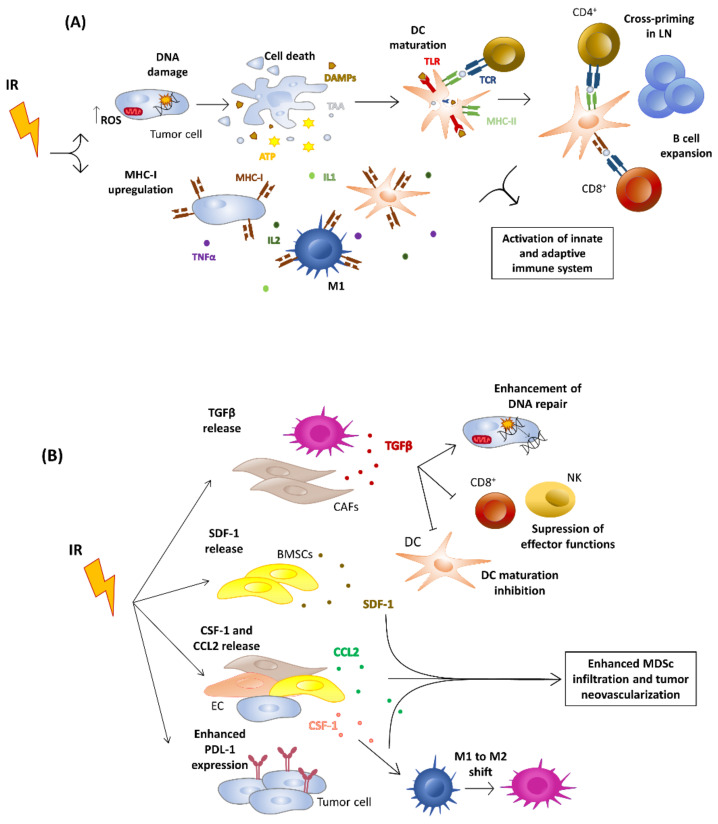
An overview of the Antitumorigenic and Pro-Tumorigenic effects of Radiotherapy. Basis of radiation-induced immunogenic cell death (**A**). Irradiation induces an increase in both ROS and MHC-I molecules in the tumor microenvironment, leading to DNA damage and more permissive antitumor activity by enhancing innate and adaptive cell activation. DNA damage induces apoptosis in tumor cells with the consequent release of ATP, DAMP and TAA which, through interaction with TLRs in DCs, cause their maturation. These mature DCs will migrate to the LN where they will cross-present Ag through MHC-I and II to CD8 and CD4 T cells, respectively, and trigger the clonal expansion of B lymphocytes. Barriers to radiation-induced immunogenic cell death (**B**). Irradiation can also induce the release of immunosuppressive molecules: TGFβ release can inhibit DC maturation, suppress immune effector functions and enhance DNA repair; SDF-1, CSF-1 and CCL2 attract myeloid-derived suppressor cells and induce neovascularization; PDL-1 on the tumor surface can interact with its PD-1 counterpart on T cells, leading to their inactivation. Point arrows mean activation while blunt arrows mean inhibition. ROS reactive oxygen species; DC, dendritic cell; DAMPs, damage-associated molecular patterns; TAA, tumor associated antigens; TLR, toll-like receptor; TCR, T cell receptor; MHC, major histocompatibility complex; TNFα, tumor necrosis factor; IL, interleukin; LN, lymph node; BMSCs, born marrow stromal cells; CAFs, cancer associated fibroblast; TGFβ, transforming growth factor receptor; NK, natural killers; SDF-1 stromal cell-derived factor 1; CCL2, C-C motif chemokine ligand 2; CSF-1, colony-stimulating factor 1; PDL-1, programmed death ligand 1; MDSc, myeloid derived suppressor cells.

**Figure 2 cancers-15-00021-f002:**
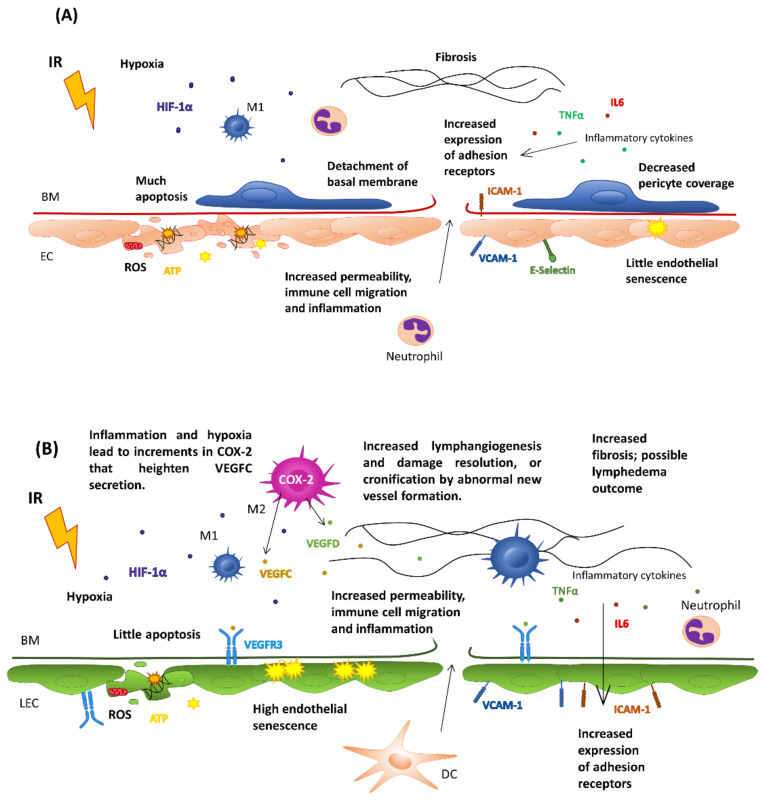
Effects of Ionizing Radiation on the Blood (**A**) and Lymphatic vasculature (**B**). IR, irradiation; HIF-1α, hypoxia-inducible factor 1; M1, macrophage type 1; M2, macrophage type 2; BM, basal membrane; EC, endothelial cell; LEC, lymphatic endothelial cell; ROS, radical oxygen species; ICAM-1, intercellular adhesion molecule 1; V-CAM, vascular cell adhesion molecule 1; TNFα, necrosis tumoral alfa; IL, interleukin; ATP, adenosine triphosphate; VEGFC, vascular endothelial growth factor C; VEGFD, vascular endothelial growth factor D; COX2, cyclooxygenase-2; DC, dendritic cell.

**Figure 3 cancers-15-00021-f003:**
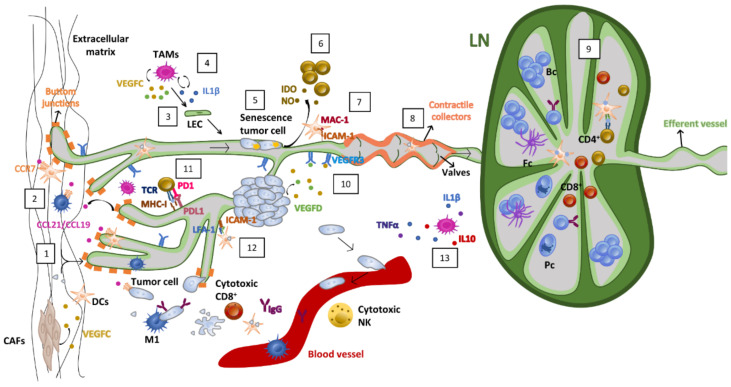
The Ying and Yang of the Lymphatic Vasculature in the Context of Tumor Microenvironment. Whereas the lymphatic system is essential in maintenance of internal homeostasis, it could also favor tumor cell metastasis: 1. Fluid and particle absorption through lymphatic capillaries; 2. Chemoattractant migration gradient of immune and tumoral cells driven by CCR7-CCL21/CCl19 interaction; 3. Macrophage transdifferentiation into LECs; 4. IL1β and VEGFC induce LEC proliferation and migration; 5. Tumor cells can acquire a senescence state while migrating through the lymphatic system; 6. IDO and NO factors released from LECs can lead to T cell proliferation inhibition; 7. MAC-1 and ICAM-1 interaction induce DC maturation inhibition; 8. Lymph pressure, together with muscles and valves on contractile collectors, impulse cells into LN; 9. B cell maturation and activation takes place on LN together with T cell activation, expansion and migration; 10. VEGFC/D ligands released by some tumoral, fibroblastic and immune cells, generate an increment in permeability, lymphangiogenesis and metastasis while interacting with VEGFR3 receptors on lymphatic vessels’ surfaces; 11. LECs can express PDL1 on each surface, playing a fundamental role in peripheral tolerance balance; 12. Leukocyte migration under inflammation is driven by LFA-1 and ICAM-1 interaction; 13. Some inflammatory cytokines such as IL1β, TNFα and IL10 can attract VEGFC-secreting leukocytes. TAMs, tumor associated macrophages; VEGFC/D, vascular endothelial growth factor C/D; LEC, lymphatic endothelial cell; IDO, indoleamine-pyrrole 2,3-dioxygenase; NO, nitric oxide; MAC-1, macrophage-1 antigen; ICAM-1, intercellular adhesion molecule 1; VEGFR3, vascular endothelial growth factor receptor 3; LN, lymph node; Bc, B cells; Fc, follicular cells; Pc, plasma cells; TNFα, tumor necrosis factor; IL1β, interleukin 1β; IL10, interleukin 10; NK, natural killer cell; LFA-1, lymphocyte function-associated antigen-1; TCR, T-cell receptor; MHC-I, major histocompatibility complex I; PD1/PDL1, programmed death-1/programmed death ligand-1; CCR7, C-C chemokine receptor type 7; CCL21/19, C-C chemokine ligand 21/19; DCs, dendritic cells; CAFs, cancer associated fibroblasts.

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
