# Peer review of "The Lymphatic Endothelium in the Context of Radioimmuno-Oncology"

_cancers, 2022, doi:10.3390/cancers15010021_

Round 1
Reviewer 1 Report
“As a first-line treatment, RT is known to modulate the immune microenvironment,”
Comment: not only, RT is given after surgery to eradicate remnant tumoral tissues, tumoral cell in transit in the lymphatic vessels and in the lymph nodes
“ but it is still unknown whether tumor lymphatic vessels influence RT efficacy.”
“Conversely, RT affects lymphatic vasculature through the development of lymphedema, compromising their viability and tissue homeostasis.”
Comment: “RT affects lymphatic vasculature” YES but lymphedema is the consequence and lymphedema by itself is not compromising the lymphatic vessels
“we now know that 49 not all the patient’s benefit equally from radiotherapy.”
Comment; sensitivity to irradiation differ from subject to subject
“Mechanisms to restore radiation damage appear in tumors as they evolve, making them resistant to radiotherapy such as the upregulation of PDL-1 in tumor and immune cells…”
Comment: we disagree. Resistance to radiotherapy is not related to biological modifications as stated by the authors
“Immunotherapy has revolutionized cancer treatment and is considered an integral piece of multimodal therapies [4].”
Comment: immunotherapy may be used for antibody based therapy and/or for biological therapy affecting the immune responses. Clarify.
General comment; lymphatic vessels are “radio-resistant” but regenerating lymphatic vessels thorough a surgical plane are “radio-sensiitve”
“The risk of developing lymphedema increments exponentially with the dosage and distance of the irradiated organ to the lymph nodes.”
Comment: “with the distance”, not clear. Clarify.
“This debilitating condition can damage lymphatic vessels”
Comment: lymphedema does not damage lymphatic vessels but is the consequence of damaged lymphatic vessels
“In breast cancer, lymphedema affects 1 in 4 patients who undergo RT. As reviewed by Allam et al., RT has also been found to increase the risk of lower extremity lymphedema by up to 40% in patients with gynecologic cancers.”… “Therefore, it is not surprising that radiation therapy is one of the most common causes of lymphatic damage.”
Comment: if we agree with the first sentences, we disagree with the sentence “Therefore, it is not surprising that radiation therapy is one of the most common causes of lymphatic damage”.
General comment: patients who develop breast cancer related upper limb lymphedema seem to present genetic and patho-physiological lymphatic predispositions
Author Response
Response to REVIEWER 1
Thank you very much for your comments. Here, we provide a point by point response to each question raised. We hope they properly clarify the issues raised and improve the previous version. As a result, we have modified the text included in the Abstract, the Introduction and the section number 6: section 6: “Effects of Radiotherapy on the Lymphatic Endothelium. Lymphedema ”, to improve clarity.
1-“As a first-line treatment, RT is known to modulate the immune microenvironment….”
Comment: not only, RT is given after surgery to eradicate remnant tumoral tissues, tumoral cell in transit in the lymphatic vessels and in the lymph nodes
Response: We agree with this first comment and improved the meaning of the sentence. We have modified the whole abstract to accommodate this broader point of view.
“The study of lymphatic tumor vasculature has been gaining interest in the context of immunotherapy of cancer. These vessels constitute conduits for immune cells' transit toward the lymph nodes and endow tumors with routes to metastasize to the lymph nodes and from them toward distant sites. In addition, this vasculature participates in the modulation of the immune response directly through the interaction with tumor-infiltrating leukocytes and indirectly through the secretion of cytokines and chemokines that attract leukocytes and tumor cells. Radiotherapy constitutes the therapeutic option for more than 50% of solid tumors. Besides impacting on transformed cells, RT affects stromal cells such as endothelial and immune cells. Mature lymphatic endothelial cells are resistant to RT, but we don’t know to what extent RT may affect tumor aberrant lymphatics. RT compromise lymphatic integrity and functionality and is a risk factor to the onset of lymphedema, a condition characterized by deficient lymphatic drainage and compromised tissue homeostasis. This review aims to provide evidence of RT's effects on tumor vessels, particularly on lymphatic endothelial cell physiology and immune properties. We will also explore the therapeutic options available so far to modulate signaling through lymphatic endothelial cell receptors and their repercussions on tumor immune cells in the context of cancer. There is a need for careful consideration of the RT dosage to come to terms with the participation of the lymphatic vasculature in anti-tumor response. Here, we provide new approaches to enhance the contribution of the lymphatic endothelium to Radio-Immune Oncology”
2-“… but, it is still unknown whether tumor lymphatic vessels influence RT efficacy.”
Response: following referee 1 comments, we have improved the redaction (see above) .
3-“Conversely, RT affects lymphatic vasculature through the development of lymphedema, compromising their viability and tissue homeostasis.”
Comment: “RT affects lymphatic vasculature” YES but lymphedema is the consequence and lymphedema by itself is not compromising the lymphatic vessels.
Response: Thank you very much for this comment, we have modified the redaction for the sake of clarity.
“RT compromise lymphatic integrity and functionality and is a risk factor to the onset of lymphedema, a condition characterized by deficient lymphatic drainage and compromised tissue homeostasis.”.
4-“we now know that not all the patient’s benefit equally from radiotherapy”
Comment; sensitivity to irradiation differs from subject to subject.
Response: thank you, this comment, again will help us to improve clarity. Below you will find the modified sentence and some considerations we have added just for the reviewer.
…”sensitivity to irradiation differs from subject to subject: radiation dosage, fractioning irradiated tissue and volume of irradiation. In addition, there are some specific individual factors such as age, sex, life style or genetics and epigenetics affecting radiosensitivity [1]. “
For the reviewer:
Nevertheless, a polygenic model of radiosensitivity is the most accepted now a days, being supported by recent High-Throughput Genetic Screening techniques including genome wide association studies, microarrays or NGS among others (https://journals.sagepub.com/doi/10.1177/0146645318764091; https://www.ncbi.nlm.nih.gov/pmc/articles/PMC9369104/ )
5-“Mechanisms to restore radiation damage appear in tumors as they evolve, making them resistant to radiotherapy such as the upregulation of PDL-1 in tumor and immune cells…”
Comment: we disagree. Resistance to radiotherapy is not related to biological modifications as stated by the authors
Response: Our aim with this sentence was to discuss the importance of radiotherapy in the modulation of the anti-tumor response. To clarify this point, we have added some information to put this issue in context and modify sentence order to avoid confusion. An additional reference has been introduced to support this new information.
“Radiotherapy is a technique based on the use of high-energy accelerated particles to damage the DNA of the irradiated cells, preventing their replication and causing their death. It has been used for more than a century as a treatment for cancer to eradicate tumors, reduce recurrence or as palliative treatment. Nevertheless, sensitivity to irradiation differs from subject to subject: radiation dosage, fractioning irradiated tissue and volume of irradiation. In addition, there are some specific individual factors such as age, sex, life style or genetics and epigenetics affecting radiosensitivity [1].
Mechanisms to restore radiation damage appear in tumors as they evolve, making them resistant to radiotherapy. These mechanisms are multifactorial and may be a result of intrinsic conditions such as alterations in the DNA repair machinery, or emanate from tumor microenvironmental conditions such as hypoxia [2]. The tumor microenvironment plays a prominent role in counteracting the effects of radiation. For example, dysfunctional endothelial vessels impair oxygen access to the tumor and therefore lessen the amount of radical oxygen species produced as a consequence of radiation [3]. This hypoxic state promotes further myeloid-derived suppressor cells (MDSC) infiltration. Importantly, defective vasculature impairs immune cell migration into the affected areas to phagocyte radiation-induced tumor antigens and present them in the lymph nodes or cross-present them to tumor cross-presenting DC [4].
On the other side, the immune system constitutes part of the stromal compartment that may modulate the radiosensitivity of tumors. In fact, current treatment strategies combine radiotherapy with other therapies directed to harness stromal-derived resistances [5]. RT can induce anti-tumor immune response through the activation of the IFNg-STING pathway but, there are instances in which RT-induced immune mechanisms contribute to immune suppression through the upregulation of PDL-1 in tumor and immune cells, by the recruitment of suppressive cells and, lastly, through the induction of the secretion of proinflammatory cytokines and chemokines such as transforming growth factor β (TGFβ), chemokine CC motif ligand 2 (CCL2) or colony-stimulating factor 1 (CSF-1) among others by stromal cells [6-8].”
6-“Immunotherapy has revolutionized cancer treatment and is considered an integral piece of multimodal therapies [4].”
Comment: immunotherapy may be used for antibody-based therapy and/or for biological therapy affecting the immune responses. Clarify.
Response: We are not sure what the reviewer's specific question is in this case. In the following paragraph of the original version we detailed the various types of immunotherapies that include both the use of immunomodulatory antibodies and cell-based therapies.
“Current efforts to induce potent anti-tumor immune responses can be summarized in three categories: use of monoclonal antibodies that block or activate immune-signaling receptors, use of vaccines against tumor-specific antigens or introduction of tumor-directed vectors that express immunomodulatory cytokines and lastly, the administration of immune cells engineered to strongly and specifically recognize tumor antigens to mount a comprehensive immune-response.”
7-General comment; lymphatic vessels are “radio-resistant” but regenerating lymphatic vessels through a surgical plane are “radio-sensitive”
Response: We agree with the reviewer that proliferative vessels are more susceptible to RT than quiescent cells. In fact, the differential effect of RT on mature and immature vessels has been illustrated by several studies (https://pubmed.ncbi.nlm.nih.gov/9043022/). The explanation under this phenomenon is related to the proliferative capacity exerted by endothelial or any other cell. Hence, cells on G2/M phase present higher radiosensitivity than those in G1 or S phase (https://pubmed.ncbi.nlm.nih.gov/15234026/#:~:text=The%20cell%20cycle%20phase%20also,part%20of%20the%20S%20phase ).
Therefore, although mature lymphatic vessels may be more resistant to radiotherapy than its blood counterparts, as it is stated in the text, proliferative lymphangiogenic or regenerating lymphatic vessels seem to be sensitive to ionizing radiation. In fact, it has been described how the lymphangiogenic factor VEGF-C enhances the radiosensitivity of lymphatic endothelial cells (https://pubmed.ncbi.nlm.nih.gov/24201897/ ).
To make a more understandable version of this section we have reorganized some of the paragraphs an included new references. The corrected version of the section 6, as appears in the corrected version of the manuscript goes:
“In the clinic, the lymphatic vessels are often included in the irradiated field and suffer the consequences of ionizing radiation, albeit at a different magnitude than their blood counterparts. Being the conduits in charge of the transit of antigen-presenting cells from the tumor to the lymph nodes to mount the anti-tumor response, it is surprising that relatively few studies address how ionizing radiation affects its structure and functionality.
Although the lymphatic vasculature is not covered by pericytes and present intermittent adhesion structures, they are suggested to be resistant to radiotherapy than blood vessels. Several experiments support this notion. For example, in an analysis of the lymphatic vasculature in the skin biopsies from breast cancer patients, the total amount of total lymphatic vessels was similar in irradiated and non-irradiated tissues. However, irradiated tissues presented lower numbers of small caliber vessels while there was an increment in the total amount of high-caliber lymphatics. Loss of the lymphatic microvasculature in the irradiated field was associated with higher of TGFβ levels in the irradiated tissue, while VEGF-C secretion by tumor resident macrophages was suggested to be associated with the increments observed in high-caliber vessels [117]. In addition it was demonstrated in mouse how lymphatic endothelial cells of the small intestine and peri-tumoral areas were more resistant to radiation injury than blood vessels [148].
As it occurred with other tissues the damaged infringed on the lymphatic vasculature depends on radiation dosage. In mouse models for lung cancer that received ionizing radiation, it was demonstrated how radiated A549 and H1299 NSCLC cells secreted VEGFC in a dose-dependent manner that resulted in increase in MLVD in irradiated tumors with five fractions of 5 Gy when compared to non-irradiated mice. The same vessels present severe necrosis when mice were irradiated with five fractions of 10 Gy. [149]. Proliferative vessels, such those present in the tumors, are by nature more sensitive to irradiation than stable adult vasculature. For example, most tumors, secrete lymphangiogenic factors that at least in in vitro, the lymphangiogenic factor VEGF-C enhances the radiosensitivity of LEC [150]
In a study of high dose irradiation of mouse lungs, Cui et al reported a significant decrease in lymphatic vessels associated with radiation exposure in spite of presenting higher amounts of VEGFC and VEGFD positive alveolar macrophages. In addition, the fibrotic lesions in irradiated mouse lungs exhibited strong immunoreactivity for VEGFC [112].
8-“The risk of developing lymphedema increases exponentially with the dosage and distance of the irradiated organ to the lymph nodes.”
Comment: “with the distance”, not clear. Clarify.
Response: thank you for your observation. We have modify the sentence to improve its reading: .
“There are two types of lymphedema: primary and secondary. Primary lymphoedema result from abnormal development and/or functioning of the lymphatic system. Mutations in numerous genes involved in the initial formation of lymphatic vessels (including valves) as well as in the growth and expansion of the lymphatic system are identified in y about one-third of affected individuals [157]. Alterations of the VEGFR3 signaling pathway are a common feature of primary lymphedema, and supplementation with VEGFC-releasing patches or administration of HGF or ANG-2 improve lymphedema condition, at least in animal models [158]. Secondary lymphedema, however can be triggered in response to surgery or irradiation of lymphatic vessels. Unlike lymphatic vessels, lymph nodes are highly radiosensitive due to the fatty-acid change and fibrosis experienced after being irradiated [159]. Therefore, the risk of developing lymphedema might be higher if lymph nodes have been resected or if the tumor has few or many nearby lymph nodes [160]. The risk of developing lymphedema increases exponentially with RT dosage and distance of the lymph node to the irradiated organ: the closest LNs to the irradiated, the higher the affectation [161].”
9-“This debilitating condition can damage lymphatic vessels”
Comment: lymphedema does not damage lymphatic vessels but is the consequence of damaged lymphatic vessels.
Response: We have changed the sentence to make clear the point of lymphedema being the consequence of damaged lymphatic vessels but not its root:
“Lymphedema is a common condition in patients with breast cancer and is characterized by a malfunction and destabilization of the lymphatic vessels. This debilitating condition appears as a consequence of damaged lymphatic vessels that lead to edema, fibrosis, inflammation, and dysregulated adipogenesis. In consequence, there is marked swelling of the affected limb with all the associated morbidities.”
10-“In breast cancer, lymphedema affects 1 in 4 patients who undergo RT. As reviewed by Allam et al., RT has also been found to increase the risk of lower extremity lymphedema by up to 40% in patients with gynecological cancers.”… “Therefore, it is not surprising that radiation therapy is one of the most common causes of lymphatic damage.”
Comment: if we agree with the first sentences, we disagree with the sentence “Therefore, it is not surprising that radiation therapy is one of the most common causes of lymphatic damage”.
General comment: patients who develop breast cancer related upper limb lymphedema seem to present genetic and patho-physiological lymphatic predispositions.
Response: Thank you very much for your observation, we have added more information that addresses the reviewer’s concerns in the following sentence.
In breast cancer, lymphedema affects 1 in 4 patients who undergo RT. As reviewed by Allam et al., RT increase the risk of lower extremity lymphedema by up to 40% in patients with gynecologic cancers. Furthermore, the RT delivery technique also affects lower extremity lymphedema rates [162]. In consequence, radiation therapy is one of the risk factors behind lymphatic damage adding to genetic [163,164] and patho-physiological factors such as partial lymphatic resection during cancer surgery, infections [165], inflammation [166], venous diseases or obesity [167].
Reviewer 2 Report
This is an exceptionally good review and balanced assessment of the status of RT's effects on tumor lymphatic endothelial cell physiology and immune properties. The article highlights important data that might have been overlooked when promulgating the clinical value of cancer immunotherapy and related trials. It will provide novel approaches to enhance the contribution of the lymphatic endothelium to Radio-Immune Oncology. The manuscript can be improved in the following points.
1. An emerging role of HIF biology is its effects on the tumor microenvironment. The EGLN/HIF axis plays a key role in regulating the function of the various components of the tumor microenvironment, which include cancer-associated endothelial cells, immune cells, and the extracellular matrix (ECM).
2. The importance of late effects of radiation increases, particularly those caused by vascular endothelial injury. Radiation both initiates and accelerates atherosclerosis, leading to vascular events like stroke, coronary artery disease, and peripheral artery disease. Increased levels of proinflammatory cytokines in the blood of long-term survivors of the atomic bomb suggest that radiation evokes a systemic inflammatory state responsible for chronic vascular side effects.
3. Tumor cells are known to induce platelet activation and endothelial dysfunction in cancer metastasis. They have also been implicated to promote tumor metastasis through platelet-tumor cell interactions. Platelet-tumor cell interactions promote tumor cell survival and dissemination in blood circulation.
Author Response
This is an exceptionally good review and balanced assessment of the status of RT's effects on tumor lymphatic endothelial cell physiology and immune properties. The article highlights important data that might have been overlooked when promulgating the clinical value of cancer immunotherapy and related trials. It will provide novel approaches to enhance the contribution of the lymphatic endothelium to Radio-Immune Oncology.
Response: Thank you for your positive review on the paper and the suggestions you mention. Although all of them dwell on important aspects of cancer biology, we have incorporated those related to hypoxia and atherosclerosis but omitted the issue on platelets. We feel that this last aspect raises an important aspect of the metastatic process but it deviates from the scope of the present review.
- An emerging role of HIF biology is its effects on the tumor microenvironment. The EGLN/HIF axis plays a key role in regulating the function of the various components of the tumor microenvironment, which include cancer-associated endothelial cells, immune cells, and the extracellular matrix (ECM).
Response. We agree with the reviewer in the convenience of discussing the role of hypoxia in the response to radiotherapy. To that aim we have included the following paragraph
“Hypoxia, limits the sensitivity of tumors to radiation, which preferentially kills well-oxygenated cells. In fact, cells irradiated in the absence of oxygen are 2- to 3-fold more radioresistant than well-oxygenated cells (DOI:10.1016/j.hoc.2006.01.007). The cellular adaptation of the tumor to hypoxia is driven by the EGLN/HIF-1 axis that is active in different components of the tumor microenvironment (DOI: 10.1007/s11427-017-9178-y) including tumor endothelial cells. In fact, HIF-1 activation post-irradiation protects the vasculature countering the oxidative stress caused by irradiation leading to maintained tumor-vessel integrity and tumor perfusion (: 10.1158/1078-0432.CCR-12-0858). Interestingly, recently it has been published how targeting of HIF-1a post radiotherapy improves anti-tumor immunotherapy and the efficacy of radiotherapy (10.3390/cancers14133273)”.
- The importance of late effects of radiation increases, particularly those caused by vascular endothelial injury. Radiation both initiates and accelerates atherosclerosis, leading to vascular events like stroke, coronary artery disease, and peripheral artery disease. Increased levels of proinflammatory cytokines in the blood of long-term survivors of the atomic bomb suggest that radiation evokes a systemic inflammatory state responsible for chronic vascular side effects.
Response: We appreciate the comment and considered adding some sentences in which the long term effects of RT in arterial vasculature. The modified sentence will be:
“Vascular endothelial injury is an important component of late radiation-induced morbidity, and affects several tissues and organs, including the skin [155], kidney (https://pubmed.ncbi.nlm.nih.gov/11845960/ ), lung [140], bowel (https://pubmed.ncbi.nlm.nih.gov/10719695/) and heart. In fact, coronary diseases such as stroke could be prompted due to thrombosis and atherosclerosis initialization and acceleration post-radiation treatment (https://pubmed.ncbi.nlm.nih.gov/30175280/; https://www.ncbi.nlm.nih.gov/pmc/articles/PMC6223160/ )
- Tumor cells are known to induce platelet activation and endothelial dysfunction in cancer metastasis. They have also been implicated to promote tumor metastasis through platelet-tumor cell interactions. Platelet-tumor cell interactions promote tumor cell survival and dissemination in blood circulation.
Response: After discussing the comment and studying the subject, we feel that the study of the participation of platelets in the process of metastases is out of the scope of the present review for the following reasons: on the one hand, we are not focusing in the process of metastasis but in the effects of radiation on the tumor niche in the context of immunotherapy and, secondly, as our main subject is the lymphatic vasculature that is deprived of platelets, we think they are not relevant to the biology of tumor lymphatics.
Reviewer 3 Report
Suarez et al., described in there review the role radiotherapy and immunotherapy to modulate lymphatic function the use in therapy.
This is a very interesting and well-written review. I really enjoyed reading this article.
There is only a very minor point of criticism: The figures are well-done however, some parts are very small and the figures are somehow overloaded (in particular Fig.3). In addition the authors used very light colours (yellow, orange) that is sometimes not easy to distinguish.
Author Response
Suarez et al., described in their review the role of radiotherapy and immunotherapy to modulate lymphatic function the use in therapy.
This is a very interesting and well-written review. I really enjoyed reading this article.
Response: We appreciate this positive review and tried to improve the clarity of the figures as suggested by the reviewer.
1-There is only a very minor point of criticism: The figures are well-done however, some parts are very small and the figures are somehow overloaded (in particular Fig.3). In addition the authors used very light colors (yellow, orange) that are sometimes not easy to distinguish.
Response: Thank you very much for your advice and the useful feedback. We have reviewed all the figures and changed the lighter colors to darker ones to ensure image clarity. In addition, we have eliminated the text on figure 3 and added it to the corresponding figure legend.